# Epitaxial Growth and Structural Characterizations of MnBi_2_Te_4_ Thin Films in Nanoscale

**DOI:** 10.3390/nano11123322

**Published:** 2021-12-07

**Authors:** Shu-Hsuan Su, Jen-Te Chang, Pei-Yu Chuang, Ming-Chieh Tsai, Yu-Wei Peng, Min Kai Lee, Cheng-Maw Cheng, Jung-Chung Andrew Huang

**Affiliations:** 1Department of Physics, National Cheng Kung University, Tainan 701, Taiwan; macg0510@yahoo.com.tw (S.-H.S.); c44051079@gs.ncku.edu.tw (J.-T.C.); chuang.py@nsrrc.org.tw (P.-Y.C.); maggie089621@gmaIl.com (M.-C.T.); g73582@gmail.com (Y.-W.P.); anion3143@hotmail.com (M.K.L.); 2National Synchrotron Radiation Research Center, Hsinchu 300, Taiwan; 3Department of Physics, National Sun Yat-sen University, Kaohsiung 80424, Taiwan; 4Graduate Institute of Applied Science and Technology, National Taiwan University of Science and Technology, Taipei 106335, Taiwan; 5Taiwan Consortium of Emergent Crystalline Materials, Ministry of Science and Technology, Taipei 10601, Taiwan; 6Department of Applied Physics, National University of Kaohsiung, Kaohsiung, 811, Taiwan

**Keywords:** MnBi_2_Te_4_, topological insulators, antiferromagnetic order, molecular beam epitaxy

## Abstract

The intrinsic magnetic topological insulator MnBi_2_Te_4_ has attracted much attention due to its special magnetic and topological properties. To date, most reports have focused on bulk or flake samples. For material integration and device applications, the epitaxial growth of MnBi_2_Te_4_ film in nanoscale is more important but challenging. Here, we report the growth of self-regulated MnBi_2_Te_4_ films by the molecular beam epitaxy. By tuning the substrate temperature to the optimal temperature for the growth surface, the stoichiometry of MnBi_2_Te_4_ becomes sensitive to the Mn/Bi flux ratio. Excessive and deficient Mn resulted in the formation of a MnTe and Bi_2_Te_3_ phase, respectively. The magnetic measurement of the 7 SL MnBi_2_Te_4_ film probed by the superconducting quantum interference device (SQUID) shows that the antiferromagnetic order occurring at the Néel temperature 22 K is accompanied by an anomalous magnetic hysteresis loop along the *c*-axis. The band structure measured by angle-resolved photoemission spectroscopy (ARPES) at 80 K reveals a Dirac-like surface state, which indicates that MnBi_2_Te_4_ has topological insulator properties in the paramagnetic phase. Our work demonstrates the key growth parameters for the design and optimization of the synthesis of nanoscale MnBi_2_Te_4_ films, which are of great significance for fundamental research and device applications involving antiferromagnetic topological insulators.

## 1. Introduction

Magnetic topological insulators (TIs) are an attractive platform because their finite magnetic moment provides mass to the massless Dirac fermions, thereby opening an energy gap in an otherwise gapless Dirac state, leading to several emerging topologically driven quantum states [1,2,3]. Two salient examples are the quantum anomalous Hall (QAH) insulator [4,5] and the axion insulator states [6,7,8]. These axion insulator states require an atypical magnetic profile; the magnetizations of the top and bottom surfaces of the TI must align oppositely [6,7,8,9,10]. Axion insulators have been observed so far only in atomically tailored TI heterostructures. A natural axion insulator was discovered in layered compound MnBi_2_Te_4_, in which the necessary magnetic profile was achieved through an intrinsic antiferromagnetic coupling of the adjacent atomic layers. Furthermore, upon applying a magnetic field to control the arrangement of the interlayer spins in MnBi_2_Te_4_, the states of QAH insulators and axion insulators become interchanged [11,12,13,14,15,16,17,18,19,20]. Axion insulators and Chern insulators have recently been realized in mechanically exfoliated MnBi_2_Te_4_ flakes [12,21,22,23], but such flakes are fragile and have small and irregular shapes, thus preventing the large-scale fabrication of devices that utilize the QAH effect. Moreover, the fabrication of MnBi_2_Te_4_ flakes requires heating to a high temperature, which readily causes high-density intrinsic defects. Such defects in MnBi_2_Te_4_ not only cause bulk metallic conductivity, preventing the measurement of the quantum transport of the surface states, but it might also affect magnetic and topological properties [24,25].

In these respects, the molecular beam epitaxy (MBE) method is potentially a promising technique for the synthesis of van der Waals (vdW) materials such as MnBi_2_Te_4_ because the deposition temperature is lower than for flakes grown in thermal equilibrium; the former can decrease the defect concentration [26,27]. Furthermore, the thin-film configuration can provide a suitable platform to realize various artificial MnBi_2_Te_4_ stacking structures, for example, heterostructures and hybrid systems with other materials such as ferromagnets [28,29,30,31,32]. Hence, the MBE growth of high-quality MnBi_2_Te_4_ films is essential for the practical development of devices specifically designed to exploit the remarkable properties of the QAH state and axion insulator state. To date, the MBE growth of MnBi_2_Te_4_ has been pursued by a few groups [31,32,33,34,35,36,37,38]. Despite great effort, the preparation of high-quality MnBi_2_Te_4_ thin films has become a major challenge in this developing field.

In this work, we report the synthesis of MnBi_2_Te_4_ films grown on sapphire substrates with the MBE method. We demonstrate that a self-regulated growth window at the optimal substrate temperature controlled with a Mn/Bi flux ratio emerged through the higher volatility of Te and yielded high-quality MnBi_2_Te_4_ films of a single phase. The MBE growth parameters and structural characterization of the MnBi_2_Te_4_ films were studied with X-ray diffraction (XRD) and Raman spectra. The magnetic properties of MnBi_2_Te_4_ thin films characterized with SQUID indicate that antiferromagnetic order occurring at Néel temperature 22 K was accompanied by an anomalous magnetic hysteresis loop along the *c*-axis. The band structure studied with angle-resolved photoemission spectra revealed a Dirac-like surface state. This work establishes a systematic approach to the epitaxial growth of high-quality MnBi_2_Te_4_ films for future device applications.

## 2. Experimental Section

The MnBi_2_Te_4_ samples were grown on *c*-plane Al_2_O_3_ substrates with the molecular beam epitaxy (AdNaNo Corp., model MBE-9, New Taipei, Taiwan) method in an ultrahigh vacuum (UHV) chamber equipped with reflection high-energy electron diffraction (RHEED) [39,40]. Highly pure Mn (99.99%), Bi (99.9999%) and Te (99.9999%) were co-evaporated from Knudsen effusion cells. The cell temperature was adjusted precisely before growth to provide the required flux, which was calibrated with the beam flux monitor. Before loading Al_2_O_3_ substrates into the growth chamber, the Al_2_O_3_ substrates were ultrasonically cleaned in acetone, isopropyl alcohol and deionized water for 10 min, respectively, and then blown by pure N_2_ to the surface before being loaded into the growth chamber. To remove possible contaminants on the surface, the Al_2_O_3_ substrates were heated to 1000 °C and kept for 1 h before the growth of the MnBi_2_Te_4_ films. During the growth, the Te flux was supplied in excess to overcome the high volatility and to decrease the formation of Te defects. The composition of the films was controlled by the Mn/Bi flux ratio. The flux ratio of Mn/Bi is calibrated with the beam flux monitor (BFM), which is an ion gauge that can measure the equivalent pressure of the molecular beam. To facilitate the epitaxial growth of MnBi_2_Te_4_, a Bi_2_Te_3_ buffer layer (1 quintuple layer, QL) was first grown on the Al_2_O_3_ substrate at a substrate temperature of 370 °C. The Bi_2_Te_3_ layer was further annealed to the desired growth temperatures under Te-rich conditions; the MnBi_2_Te_4_ films were then deposited on the buffer layers. The nominal growth rate of the Bi_2_Te_3_ and MnBi_2_Te_4_ films were QL/7 min and SL/4 min, respectively. The samples used for SQUID, angle-resolved photoemission spectra (ARPES) and a scanning tunneling microscope (STM) were covered in situ with amorphous Te capping layers (1~2 nm) in an MBE system to avoid surface oxidation. The crystalline condition of the grown thin film was monitored in situ with RHEED. X-ray diffraction (XRD), a high-resolution transmission electron microscope (HRTEM) and Raman spectra provided structural characterization. The Raman spectra were recorded on a micro-Raman spectrometer with an excitation wavelength of 532 nm. The elemental composition of the samples was calibrated with a transmission electron microscope energy-dispersive system (TEM-EDS) (see Appendix A). The surface morphology was characterized with an atomic force microscope (AFM). The magnetic properties as a function of temperature from 5 to 300 K were measured with a SQUID magnetometer (Quantum Design MPMS SQUID VSM system). An external magnetic field was applied to the surface of the epitaxial layer in either an in-plane or out-of-plane direction. The diamagnetic contribution of the Al_2_O_3_ substrate was determined from the slope of the magnetization M (*H*) recorded at large magnetic fields and at 300 K, well above the Néel temperature of MnBi_2_Te_4_; the derived substrate contribution was then subtracted from the raw data recorded at a lower temperature. The ARPES experiment was performed at beamline 21B1 of Taiwan Light Source in the National Synchrotron Radiation Research Center (NSRRC). Before the ARPES measurement, the Te-covered MnBi_2_Te_4_ film was annealed at 180 °C for about 1 h to remove the capping Te layer. ARPES were recorded in a UHV chamber equipped with a hemispherical analyzer (VG Scienta R4000, Uppsala, Sweden) [41]. All spectra were recorded at 80 K under base pressure 8.3 × 10^−11^ torr and at incident photon energy 24 eV. The angle resolution was about 0.2°; the overall energy resolution was better than 12 meV.

## 3. Results and Discussion

### 3.1. Structural Characterizations

To understand the structural evolution of a MnBi_2_Te_4_ film as a function of the Mn/Bi flux ratio ($\varphi_{r}$) and the growth temperature (*T*_G_), we performed X-ray diffraction (XRD) studies on samples in two series. Figure 1a shows the θ–2θ scans for which $\varphi_{r}$ was tuned from 0.09 to 1; *T*_G_ was fixed at 410 °C. All MnBi_2_Te_4_ films were *c*-axis (0001)-oriented, which was confirmed with a series of signals that closely match the (003*n*) signals of bulk MnBi_2_Te_4_ [15,42]. Additional signals were observed at about 2θ = 27° and 55°, which are consistent with the formation of a MnTe phase [34]. The intensity of the MnTe phase gradually weakened as $\varphi_{r}$ decreased from 1 to 0.1. When $\varphi_{r}$ decreased to 0.09, the MnTe phase disappeared, leaving only the MnBi_2_Te_4_ phase. It should be noted that with further decreasing$\;\varphi_{r}$, the XRD spectrum shows diffraction patterns similar to those of Bi_2_Te_3_, except that the diffraction peaks are broader than those of Bi_2_Te_3_. This suggests that at a lower Mn/Bi ratio, Mn atoms mainly act as dopants without significantly changing the crystalline structure of Bi_2_Te_3_. The trends are consistent with the previous reports [33]. Moreover, compared with the previous study [34], the quartz crystal microbalance (QCM) was used to calibrate the$\;\varphi_{r}$, which is different from the calibration by BFM in our study. The calibration with the QCM is to measure the frequency response of solid material deposited on the QCM. This $\varphi_{r}$ is often linearly related to the ratio of solid materials deposited on the substrate. The calibration method we used is to measure the partial pressure of the molecular beam (Mn, Bi, Te) in the gas phase, and this partial pressure ratio is often not so linear with the ratio of the solid materials deposited on the substrate. This is because the reaction (e.g., sorption and desorption) of gas-phase materials on the substrate can be quite complicated. Therefore, the difference in$\;\varphi_{r}$ between our work and the previous result is likely due to the different calibration methods adopted in the two systems. Figure 1b shows the XRD results for which *T*_G_ varied from 310 to 440 °C; the flux ratio was fixed at$\;\varphi_{r}$ = 0.09. For films grown at temperatures 310~340 °C, the intensities of the resolved signals were too weak, but these signals were well distinguished from those of Bi_2_Te_3_. With *T*_G_ increased to 380~410 °C, these signals became well resolved; the intensities increased, which clearly showed the (003*n*) signals of MnBi_2_Te_4_ [15,34,36,42] with no impurity phase. However, when *T*_G_ increased to 440 °C, volatile Bi-Te was desorbed and additional MnTe signals emerged because of the higher growth temperature. The RHEED in situ showed clear strip patterns, indicating a high crystalline quality and a flat surface of the MnBi_2_Te_4_ films, as shown in Figure 1d. Raman spectra were used to investigate how the change of$\;\varphi_{r}$ affected the lattice vibration and the electron–phonon interaction in the MnBi_2_Te_4_ films, as shown in Figure 1c. The characteristic phonon modes $\;A_{1g}^{2}$, $E_{g}^{2}$ and $A_{1g}$ were identified in the region of a small wavenumber. The Raman spectra of MnBi_2_Te_4_ film have an appreciable blue shift relative to Bi_2_Te_3_, which is attributed to the stronger in-plane bond of Mn-Te [42]. As$\;\varphi_{r}$ increased from 0.09 to 1, Raman spectra were observed to be red-shifted in these films. The decreasing wavenumber of the vibrational modes ($E_{g}^{2}$ and $A_{1g}$) is associated with an increase in Mn atoms until the formation of Mn interstitial clusters that might be located in the vdW gaps and connected to the Te atoms of the SL [43]. The optimal growth parameters of the MnBi_2_Te_4_ films occurred at approximately$\;\varphi_{r}\;$ = 0.09 and *T*_G_ = 410 °C. For a thin film prepared under the optimized conditions, the HRTEM was applied to examine the film quality. Figure 1e shows the HRTEM image along the *c*-axis direction, clearly showing the characteristic septuple-layer (SL) structure of the vdW stacking in a MnBi_2_Te_4_ epitaxial film. The film thickness was identified as 7 SL. The atomic structure of each SL is visible in the enlarged image (Figure 1f). As the atomic number of Bi is much greater than that of Te and Mn, the Bi atomic column looks brightest, which is also shown in the inset with the superimposed structural model (see the illustration in Figure 1e). The stoichiometry of the film was estimated using the TEM-EDS as shown in Appendix A), which showed that Mn:Bi:Te was 1:2.17:3.99, which is consistent with the chemical formula of compound MnBi_2_Te_4_. These results indicate that, for the epitaxial growth of stoichiometric crystalline MnBi_2_Te_4_ films, the growth temperature and the Mn/Bi flux ratio have narrow windows. 

### 3.2. Surface Morphology

The surface morphology depended strongly on$\;\varphi_{r}$ and *T*_G_. Figure 2a shows the surface of a Bi_2_Te_3_ film as grown, indicating flat terraces. The surface was composed of characteristic triangular terraces and steps, reflecting the growth of the hexagonal crystal Bi_2_Te_3_ along (0001) [44,45]. The individual terraces were preferentially aligned with each other; an occasional rotation of 60° was observed, possibly due to the formation of twin boundaries [44]. For$\;\varphi_{r}\;$= 0.09 and *T*_G_ = 310 °C, the film surface showed misoriented domains with ambiguous shapes and bright islands, which indicated poor crystalline quality (Figure 2b). The root mean square (rms) roughness was about 6.54 nm. Because *T*_G_ was too low, adatoms had insufficient energy to diffuse and to move to sites of least potential energy; they then formed a polycrystalline film. As *T*_G_ increased to 410 °C at$\;\varphi_{r}\;$ = 0.09, the most noticeable feature was the large and flat undefined morphology, as shown in Figure 2c. A significant reduction in 3D bright structures was observed over a large-scale area; its rms roughness was 0.82 nm, which is much less than that of Figure 2b. The layering step is distinguishable on the enlarged surface shown in Figure 2e. The step height is ~1.4 nm (inset of Figure 2e), which is the same as the height of a single SL of MnBi_2_Te_4_ [25,46]. For $\;\varphi_{r}\;$ = 0.5 and *T*_G_ = 410 °C, the surface showed randomly oriented elongated structures, which might be due to the formation of a Mn-rich phase; its rms roughness was 3.85 nm. The STM result revealed that the hexagonal atomic structure on the surface is consistent with the imaging of the topmost Te layer [46,47]; the in-plane lattice parameter was 4.0 Å, as shown in Figure 2f. These structural features confirmed by the AFM and XRD reveal the high quality of the pure MnBi_2_Te_4_ films grown by the delicate MBE method. 

### 3.3. Magnetic Properties

In the 2D limit, the odd SL of MnBi_2_Te_4_ exhibit QAH insulators and Chern insulators [14,22,48]. The 7 SL and 11SL MnBi_2_Te_4_ were used to measure the magnetic properties; these properties of a MnBi_2_Te_4_ film were inspected with a SQUID measurement. Figure 3a displays the field-cooled (FC) and zero-field-cooled (ZFC) curves of the 7 SL MnBi_2_Te_4_ film when the magnetic field (~1 T) was along the *c*-axis (*H*//*c*) and in-plane *ab* (*H*//*ab*), respectively. The overlapping FC and ZFC curves and λ-shaped feature with an inflection point indicated an A-type antiferromagnetic coupling along the *c*-axis. The Néel temperature (*T*_N_) of the sample was about 22 K, consistent with previous work [15,17,42]. The FC and ZFC curves deviated from each other when the temperature was less than *T*_N_, which implies the emergence of a net ferromagnetic moment. When the field was applied along plane *ab* (*H*//*ab*), the magnetic moment decreased to one-third, indicating that the MnBi_2_Te_4_ film had strong magnetic anisotropy. Figure 3b shows the field dependence in plane (*H*//*ab*) and out of plane (*H*//*c*) of magnetization curves of a 7 SL MnBi_2_Te_4_ sample. The *M*–*H* loop out of plane exhibits a non-linear ferromagnetic shape, whereas the *M*–*H* loop in plane is much flatter, indicating a perpendicular magnetic anisotropy with the easy magnetization axis along the *c*-axis [42,49]. For *H*//*c*, the *M*–*H* curve exhibits an anomalous magnetic hysteresis loop centered at the regime of a small magnetic field, indicating the spin-flip of individual layers and a ferromagnetic feature, which differs from the antiferromagnetic characteristics of MnBi_2_Te_4_ bulk and flakes [42,49,50]. In bulk and exfoliated flakes of MnBi_2_Te_4_, the *M*–*H* curves are nominally flat in the regime of a small magnetic field. The residual ferromagnetic response might originate from a possible substrate-induced effect, asymmetric upper and lower surfaces, or uncompensated layers [17,48]. Under magnetic field 3 T, the spin-flip transition occurred as indicated by red arrows in Figure 3b. Similar large-field magnetic behavior was observed in MnBi_2_Te_4_ flakes [42,48,49,50]. Figure 3c shows the field dependence out of plane of the magnetization curve of a 7 SL MnBi_2_Te_4_ sample in the temperature range from 10 K to 30 K. The anomalous magnetic hysteresis loop dwindled with increasing temperature and disappeared at 30 K, which indicates that a magnetic phase transition from an antiferromagnetic state to a paramagnetic state occurred with increasing temperature, consistent with the observation of *M*-*T* curves in Figure 3a. Figure 3d shows the field dependence out of plane of the magnetization curves of a 7 SL MnBi_2_Te_4_ sample for varied $\;\varphi_{r}$. Compared with the sample with$\;\varphi_{r}$ = 0.09, for the sample with$\;\varphi_{r}$ = 1, the magnetic hysteresis loop out of plane vanished because of the in-plane magnetic anisotropy of the MnTe phase [51]. To study the thickness-dependent magnetism, the magnetic properties of MnBi_2_Te_4_ with 7 SL and 11 SL thickness were observed. Figure 3e shows the ZFC and FC curves of 7 SL and 11 SL MnBi_2_Te_4_ films, with the magnetic field (~1 T) along the *c*-axis (*H*//*c*). It should be noted that the Néel temperature (T_N_) of 11 SL MnBi_2_Te_4_ film slightly increases to 23 K compared to that of 7 SL. Figure 3f shows the out-of-plane *M*–*H* loop of 7 SL and 11 SL MnBi_2_Te_4_. The magnetic hysteresis loop centered at the regime of low magnetic fields is persistent in both samples. However, under high magnetic fields, the spin-flip transition of 11 SL is not obvious, which may be due to the tilt or disorder of the antiferromagnetic configuration as the thickness increases.

### 3.4. Band Structure

The electronic structure of 7 SL MnBi_2_Te_4_ with growth parameters$\;\varphi_{r}\;$ = 0.09 and *T*_G_ = 410 °C was studied with ARPES, which were recorded at 80 K, to which the sample was paramagnetically ordered. Before the ARPES measurements, an XPS experiment after removing the Te capping layer was conducted to confirm the surface condition. Mn 3*p*, Te 4*d* and Bi 5*d* features could be observed clearly from the XPS spectrum, as shown in Figure 4a. Figure 4b,c displays the band-mapping results of a MnBi_2_Te_4_ sample along direction Г-K and the second derivative of the photoemission intensity, respectively. A Dirac point of MnBi_2_Te_4_ was identified in the band gap and differed from the Dirac point buried in the valence band for Bi_2_Te_3_ [37,52]. The Fermi level was located above the Dirac point and crossed the bulk conduction band (BCB), indicating that the charge carriers were mainly *n*-type. The Fermi vector ~0.13 Å^−1^ can be identified from the momentum distribution curve (MDC). The surface carrier concentration estimated from the occupied area of the 2D FS area/the area of surface BZ without counting the spin degeneracy is around ~1.34 × 10^13^ cm^−2^, which is higher than that of previous works. [37] This observation indicates that a surface Dirac cone existed in the MnBi_2_Te_4_ film in the paramagnetic state, which is consistent with previous ARPES studies [18,37]. In addition, the band-mapping result exhibits heavily *n*-doped behavior in this work. A possible reason might be attributed to a higher substrate growth temperature in our work, which could create more Te vacancies or antisite defects. [24,25] 

## 4. Conclusions

We have presented the synthesis and characterization of MnBi_2_Te_4_ films on a nanometer scale prepared with MBE. In controlling suitable growth conditions, we demonstrated the existence of a self-regulated growth window, resulting in a single phase and the high structural quality of MnBi_2_Te_4_ films confirmed with XRD and Raman spectra. A layered structure of a large area, flat surface and vdW stacking was verified with an AFM and HRTEM. The 7 SL MnBi_2_Te_4_ film showed an antiferromagnetic characteristic with a Néel temperature of 22 K accompanying a low-field anomalous magnetic hysteresis loop along the *c*-axis. The ARPES also confirmed the existence of a Dirac surface state and revealed that the paramagnetic state of MnBi_2_Te_4_ remains a TI. It is worth pointing out that the presence of the MnTe and Bi_2_Te_3_ impurity phases will greatly reduce or destroy the topological and magnetic properties of MnBi_2_Te_4_. Understanding the synthesis of antiferromagnetic topological insulators in this series with the MBE method opens a huge direction for engineering magnetism and topological states. For example, a MnBi_2_Te_4_ film could be integrated into a heterostructure to enable a functional adjustment. Hence, antiferromagnetic TI/ferromagnetic material heterostructures with exotic quantum phases could be realized [29]. In addition, the antiferromagnetic TI/superconductor heterostructures could provide a platform to explore tunable Majorana bound states [53]. Our work paves the way for future device applications involving antiferromagnetic topological insulators.

## Figures and Tables

**Figure 1 nanomaterials-11-03322-f001:**
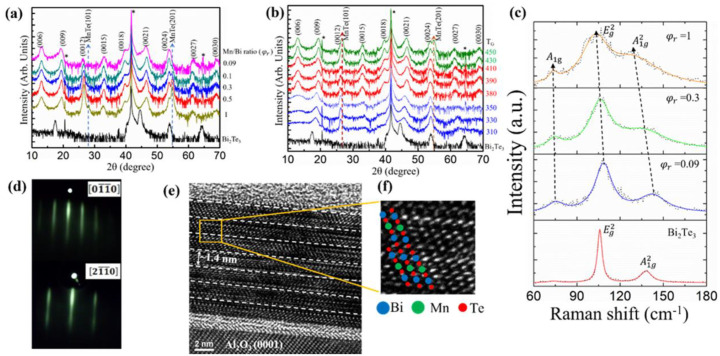
The structural characterizations of epitaxial MnBiTe films under varied growth conditions. The XRD diagrams for (**a**) *T*_G_ fixed at 410 °C with varied$\;\varphi_{r}$ and (**b**)$\;\varphi_{r}$ fixed at 0.09 under varied *T*_G_ (indicated on the right-hand axis). As indicated in the figure, the signals are marked as *: *c*-plane Al_2_O_3_ substrate. The dashed vertical arrows highlight the emergence of the MnTe phase. (**c**) Raman spectra of films for varied$\;\varphi_{r}$. The dotted lines represent raw data; the solid color lines are fits through the raw data. (**d**) RHEED patterns of films along directions $\left[0\bar{1}\bar{1}0\right]$ and $\left[2\bar{1}\bar{1}0\right]$, respectively. (**e**) TEM cross-sectional view of the film with$\;\varphi_{r}$ = 0.09 and *T*_G_ = 410 °C. (**f**) Enlarged TEM image and schematic structure of MnBi_2_Te_4_ are superimposed: blue-Bi, green-Mn, red-Te.

**Figure 2 nanomaterials-11-03322-f002:**
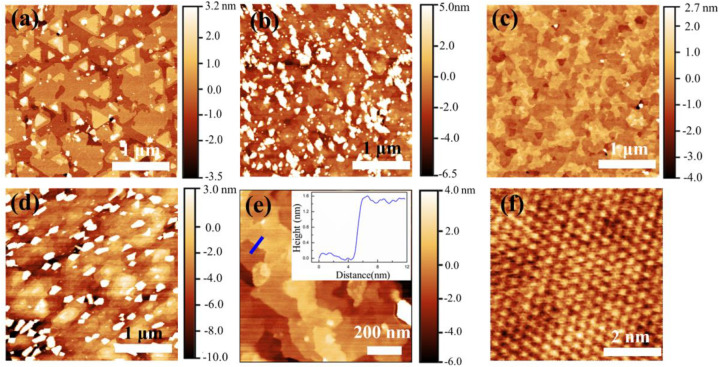
The dependence of film morphology on *T*_G_ and$\;\varphi_{r}$ from an AFM: (**a**) Bi_2_Te_3_ surface (3 μm × 3 μm). (**b**) $\varphi_{r}\;$ = 0.09 and *T*_G_ = 310 °C (3 μm × 3 μm); (**c**)$\;\varphi_{r}\;$ = 0.09 and T_G_ = 410 °C (3 μm × 3 μm); (**d**) $\;\varphi_{r}\;$ = 0.5 and *T*_G_ = 430 °C (3 μm × 3 μm); (**e**) Magnified AFM image of (**c**); the inset is a height profile along the blue solid line marked in (**e**), showing a step size of 1.4 nm. (**f**) Atom-resolved STM image of (**c**), showing Te-terminated hexagonal atomic structure (size: 6 nm × 6 nm, sample bias: 0.5 V, tunnel current: 0.2 nA).

**Figure 3 nanomaterials-11-03322-f003:**
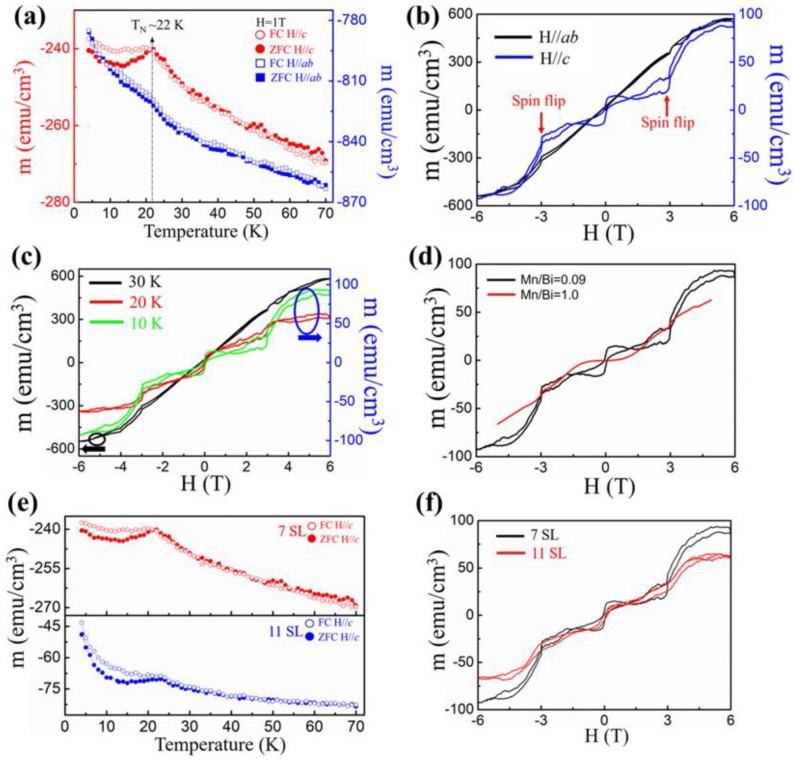
The magnetic properties of a 7 SL MnBi_2_Te_4_ film. (**a**) The temperature dependence of the magnetic moment under ZFC and FC processes with magnetic fields out of plane (*H*//*c*) and in plane (*H*//*ab*), respectively. (**b**) The field dependence of the magnetization of a film measured at 10 K with magnetic fields out of plane (*H*//*c*) and in plane (*H*//*ab*), respectively. The red arrows indicate the spin-flip transition. (**c**) The field dependence of the magnetization of a film measured at various temperatures with a magnetic field applied out of plane (*H*//*c*). (**d**) The field dependence of the magnetization of a 7 SL MnBi_2_Te_4_ film at 10 K when a magnetic field out of plane (*H*//*c*) was applied to samples with$\;\varphi_{r}$ = 0.09 and$\;\varphi_{r}\;$ = 1.0, respectively. (**e**) The temperature dependence of the magnetic moment under ZFC and FC processes with magnetic fields out of plane (*H*//*c*) for 7 SL and 11 SL, respectively. (**f**) The field dependence of the magnetization of 7 SL and 11 SL MnBi_2_Te_4_ films at 10 K when a magnetic field out of plane (*H//c*) was applied to samples.

**Figure 4 nanomaterials-11-03322-f004:**
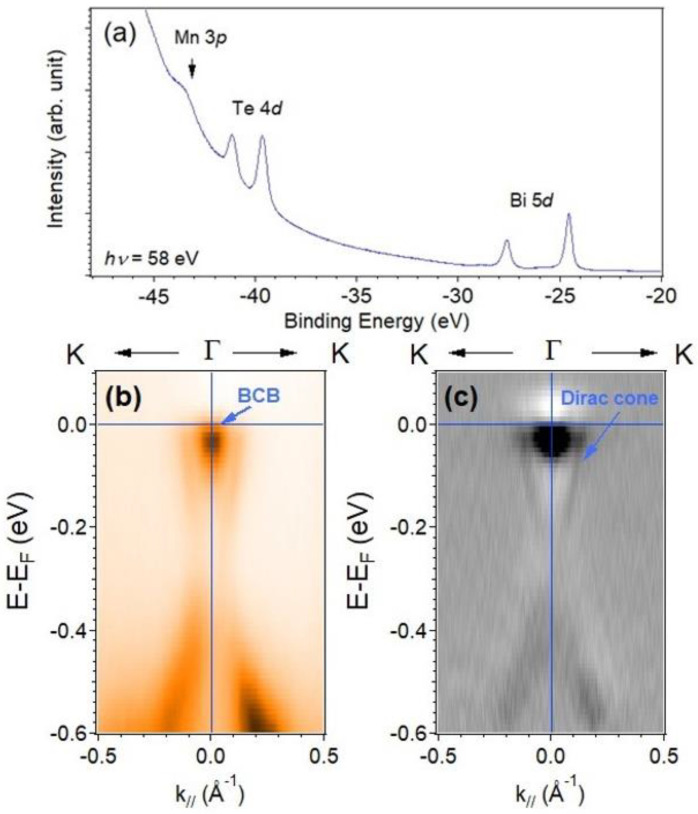
(**a**) The XPS spectrum of MnBi_2_Te_4_ with photon energy 58 eV. (**b**) The band-mapping result of a MnBi_2_Te_4_ sample measured at 80 K along direction Γ-K with photon energy 24 eV. (**c**) The second derivative of (**b**) to enhance the visibility of a Dirac cone.

## Data Availability

All data included in this study are available upon request by contact with the corresponding author.

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
