# Peer review of "Epitaxial Growth and Structural Characterizations of MnBi2Te4 Thin Films in Nanoscale"

_nanomaterials, 2021, doi:10.3390/nano11123322_

Round 1
Reviewer 1 Report
The authors of this manuscript realized the growth of topological insulator MnBi2Te4 films using MBE method. The magnetic properties and band structure of MnBi2Te4 film were also investigated. This manuscript can be accepted for publication in nanomaterials if the following questions are addressed.
- In the “experimental” part, line 76 and 110, the authors cited five published papers of their own. It is okay if the authors intended not to elaborate the methods in this manuscript. But there is no necessity here and I suspect that they are over citing their own works.
- In the “experimental” part line 80, “Al2O3 substrates were cleaned with a standard procedure” is not clear to the readers. Please show what is the standard procedure.
- Please try to improve the language, for example in line 173 “a polycrystalline film then became formed.”
- In figure 2, please include scale bar of height, which would make these AFM images much clearer and easier to read.
- The cation of figure 3 repeated the description of (d).
- I can’t see the Supplementary Materials.
Author Response
Dear Review #1:
Please see the attachment. Thanks.

Reviewer 2 Report
This manuscript reports on the MBE growth and the structure characterizations of epitaxial thin films of the antiferromagnetic topological insulator MnBi2Te4. This material is of great current interest since it has been reported that its Dirac cones are strongly influenced by the magnetic order which can be modified with an external magnetic field. The growth of high quality thin films with well controlled structural and magnetic properties has so far not been reported (to my best knowledge.
The presented results are therefore very timely and represent a great step forward toward using this highly interesting material in various kinds of multilayers and devices which utilize the magnetic field control of the topological properties.
The paper is well structured and clearly written. It starts with a concise but very useful introduction about the motivation of this work and the necessity of thin film growth of MnBi2Te4.
In the following chapter the details of the MBE growth procedure of the epitaxial films are discussed and the structural and chemical characterization with XRD, TEM, AFM, and Raman spectroscopy is presented. The presented data confirm a rather high structural quality of the films and a reasonably good control of the chemical composition.
Subsequently the papers presents magnetization measurements for the 7 SL sample. They are characteristic of an A-type AF order with an onset temperature of about 22K and a spin-flip transition around 3 Tesla that are similar as reported for bulk and exfoliated samples. Unlike the bulk samples, they observe a hysteretic response in the low-field regime that remains unexplained.
Finally, the paper presents an ARPES measurement of the electronic band structure in the vicinity of the Fermi-surface which reveals a Dirac surface state at the Gamma-point around -300 meV.
The paper concludes with a brief summary and outlook.
I find the paper generally well written and the results very interesting. I therefore recommend the paper for publication in this nanomaterials journal.
I have only a few questions and minor comments that are listed below.
- Did the authors take into account the demagnetization effects in determining the magnetic anisotropy of their thin film samples?
- The discussion of the ARPES data in Fig. 4 could be a bit more extended. It would be interesting to learn how the energy of the Dirace point around -300 meV compares to the one in bulk samples. What is the likely source of the n-type doping and how large is it as compared to the one reported for bulk samples. Are their data consistent with a lower level of defects thanks to the lower growth temperature (as it is mentioned in the growth part)?
- In the Caption of Fig. 3 the last sentence appears to be doubled.
Author Response
Dear Review #2:
Please see the attachment. Thanks.

Reviewer 3 Report
MnBi2Te4 is the first intrinsic magnetic topological insulator that exhibits intriguing properties. The authors synthesized MnBi2Te4 films using MBE. By tuning the substrate temperature and Mn/Bi flux ratio, they obtained high-quality films. The topic of the current work is of broad and timely interest to the community. The current work is important for the understanding and application of this interesting material. I would like to recommend the manuscript for publication in Nanomaterials if the following questions can be properly answered.
- The optimal Mn/Bi flux ratio of 0.09 seems too low to be true. How did the authors calibrate the flux? The optimal flux ratio also conflicts with previous results such as PRM, 4, 111201(R). The authors need to comment on this difference.
- According to previous reports, a lower Mn/Bi ratio will induce the mixture of MnBi2Te4 and MnBi4Te7. Did the authors observe the formation of MnBi4Te7 phase?
- The ARPES measured band structure is quite different from the result in CPL, 36, 076801. The topological surface state may be from the mixed Bi2Te3 since the flux ratio is too low. Although the authors argued that the observed topological surface state is not buried in the bulk states, it is similar to the topological surface state of Bi2Te3 films.
- I would like to see comparing results on films with different thicknesses.
Author Response
Dear Review #3:
Please see the attachment. Thanks.

Round 2
Reviewer 3 Report
The authors have answered most of my questions. However, I am still concerned by the extremely low flux ratio. To avoid misleading the readers, the authors should explicitly comment on the huge difference between the current work and previous results in the manuscript. After that, I will be happy to recommend the manuscript for publication in nanomaterials.
Author Response

(The authors gave the same response as above.)
